# Emergence of resistance to last-resort antimicrobials in bacteremia patients: A multicenter analysis of bloodstream pathogens in Korea

Jin Sae Yoo[1], Hui-Jin Yu[2], Kuenyoul Park[3]*, Wee Gyo Lee[4], Bo-Moon Shin[3]

1 Department of Internal Medicine, Ajou University School of Medicine, Suwon, South Korea, 2 Department of Laboratory Medicine, Seoul Medical Center, Seoul, South Korea, 3 Department of Laboratory Medicine, Sanggye Paik Hospital, School of Medicine, Inje University, Seoul, South Korea, 4 Department of Laboratory Medicine, Ajou University School of Medicine, Suwon, South Korea

☯ These authors contributed equally to this work.
* kuenyoul@paik.ac.kr

**Data Availability Statement:** The availability of the raw data described in this study is restricted by the Institutional Review Boards of three institutions (Seoul Medical Center, seoulmc.irb@gmail.com;

## Abstract

This study retrospectively reviewed the microbiological and clinical characteristics of patients diagnosed with bacteremia. Results from the first positive blood cultures were consecutively collected from July 2022 to June 2023 at a public secondary hospital, a university-affiliated tertiary hospital, and a university-affiliated secondary hospital in the Seoul metropolitan area. Antibiotic spectrum coverage (ASC) scores were calculated on the day the blood culture was performed (B0) and on two days after the blood culture results were reported (R+2). A total of 3,397 isolates were collected from 3,094 patients. Among these, 949 isolates obtained from 893 patients were classified as multidrug-resistant organisms (MDRO), including 170 imipenem-resistant gram-negative bacteria, 714 methicillin-resistant staphylococci, and 65 vancomycin-resistant enterococci. Interestingly, 13 and 42 gram-positive isolates were resistant to linezolid and quinupristin/dalfopristin, respectively. Moreover, 44 and 181 gram-negative isolates were resistant to amikacin and tigecycline, respectively. The proportion of ASC scores corresponding to broad or extremely broad-spectrum coverage was not significantly different between MDRO and non-MDRO groups at B0 (p = 0.0925). However, it increased in the MDRO group at R+2 (p <0.001). This study found that resistance to last-resort antimicrobials is emerging. Therefore, developing and incorporating molecular diagnostics using a wide range of resistance targets may facilitate rapid, tailored antimicrobial treatments.

## Introduction

Bloodstream infections (BSI) are associated with a high global disease burden. The magnitude of the burden varies depending on the BSI type (e.g., community-acquired *vs.* hospital-acquired), involved pathogens, and geographic regions. A previous review reported an

Ajou University Hospital, ajouirb@aumc.ac.kr;
Sanggye Paik Hospital, spirb@paik.ac.kr) because
it contains potentially identifiable patient
information and is therefore not publicly available.

**Funding:** The author(s) received no specific
funding for this work.

**Competing interests:** The authors have declared
that no competing interests exist.

estimated incidence of 43 to 101 per 100,000 population, with case fatality ranging from 13 to 17% [1]. BSI caused by bacteria harboring antimicrobial resistance (AMR) is particularly concerning due to greater disease burden and poorer outcomes compared to those caused by wild-type pathogens, with one study estimating that approximately 1.5 million deaths were directly attributable to BSI with AMR [2]. Bacteria harboring resistance to several classes of antimicrobial agents are called multidrug-resistant organisms (MDRO). In 2017, the World Health Organization (WHO) listed several MDRO as "priority pathogens" requiring novel therapeutic agents, including vancomycin-resistant enterococci (VRE), methicillin-resistant *Staphylococcus aureus* (MRSA), carbapenem-resistant *Acinetobacter baumannii*, carbapenem-resistant *Pseudomonas aeruginosa*, and carbapenem-resistant/extended beta-lactamase (ESBL)-producing Enterobacterales [3]. The increasing incidence of MDRO infections has prompted several guidelines on various infectious diseases to recommend empirical use of broad-spectrum antibiotics [4,5], resulting in the selection of pathogens that are resistant to these antibiotics. Conventional antimicrobial susceptibility testing is constrained by bacterial growth time, and a more rapid test that can direct appropriate antibiotic therapy is required to provide adequate antimicrobial coverage against MDRO infections and to discontinue the unnecessary administration of broad-spectrum antimicrobials as soon as possible.

Molecular diagnostics are employed more frequently to diagnose and manage BSI. Specific techniques include polymerase chain reaction (PCR), next-generation sequencing (NGS), and less commonly, fluorescence *in situ* hybridization. Notably, the impact of the BioFire blood culture identification (BCID; bioMérieux, Marcy l'Etoile, France) assay on the detection and management of AMR has been investigated in several studies because of its availability, ease of use, and relatively low cost compared with NGS. Overall, the use of BCID appears to be associated with a shorter time to optimal antibiotic therapy for BSI by 6.25 to 64 hours compared to traditional blood culture-based decisions [6–11] and optimization of antibiotic regimen in 31.8–45.1% of patients with BSI [12,13]. However, some studies have also reported no significant reduction in the time to appropriate antibiotic de-escalation or mortality rate associated with the use of BCID in patients with BSI [7,14]. In addition, the AMR genes detected by current BCID are limited to those associated with ESBL, carbapenemase production, and resistance to methicillin, vancomycin, and colistin. Recently, T2resistance (T2Biosystems, Lexington, MA, USA) has become available for research use, and one study reported an 84.6% detection rate of carbapenem resistance genes in blood culture-confirmed gram-negative BSI within 3 to 5 hours [15]. However, this method does not provide information regarding AMR to antibiotic class other than beta-lactam, carbapenem, and glycopeptide.

Such limitations of commercially available molecular AMR panels for BSI raise questions about whether they are sufficient in their breadth and accurately address "real-world" BSI landscapes. Although a large-scale analysis of antimicrobial susceptibility patterns and clinical results of BSI would be useful in answering these questions, few such studies have been reported. Therefore, this study retrospectively reviewed the microbiological and clinical characteristics of patients diagnosed with BSI at three hospitals in the metropolitan area of Seoul, Korea, to investigate the current landscape of the susceptibility profiles of BSI pathogens.

## Methods

### Study design and procedures

This retrospective observational study was conducted using the medical records and blood culture results of patients admitted to three distinct hospitals from July 2022 to June 2023: a public secondary hospital (S), university-affiliated tertiary hospital (A), and university-affiliated secondary hospital (P). All hospitalized patients with positive blood culture during the study

period were enrolled. Data were collected only for the first organism isolated from the positive blood culture of each patient's duration of admission. However, all were included if two or more organisms were simultaneously identified on the first blood culture. At the three institutions, organisms isolated from blood cultures were subjected to bacterial identification using MALDI-TOF mass spectrometry (MALDI Biotyper, Bruker Daltonics, GmbH, or Vitek MS system, bioMérieux) and Vitek2 systems (bioMérieux). Antimicrobial susceptibility testing (AST) was performed using the Vitek2 system. Bacterial identification and AST were performed in the laboratory of the patient's admitting hospital. The laboratories of all three participating hospitals were accredited for internal and external quality assurance programs by the Laboratory Medicine Foundation of Korea and Korean Association of External Quality Assessment Service during the study period. We also compiled data on the antibiotics administered during hospitalization, underlying disease of human immunodeficiency virus (HIV) infections and malignancies, history of transplantation, and in-hospital mortality rates. Underlying diseases such as HIV infection and malignancies, as well as transplantation history were searched against the main diagnosis codes registered in the electronic medical records.

## Classification of antibiotic resistance

The percent susceptibility to each antimicrobial agent was calculated for all gram-positive bacteria, VRE, methicillin-resistant staphylococci (MRS), all gram-negative bacteria, and imipenem-resistant gram-negative bacteria (IRGN). Based on their susceptibility profiles, VRE, MRS, and IRGN were classified as MDRO [16]. Antibiotics prescribed to each patient on the day of the first blood culture (B0) and on the second day after reporting the blood culture results (R+2) were recorded to evaluate the pattern of antibiotic use for empirical treatment and tailored treatment after the culture results were reported, respectively. The antibiotic prescription was analyzed only for patients who stayed in the admitted hospital on R+2 to exclude patients who died or were transferred to other hospitals before R+2. Antibiotic spectrum coverage (ASC) score, which quantifies the breadth of antimicrobial coverage provided by the prescribed antibiotics [17], was calculated for B0 and R+2. Subsequently, we assigned each patient's ASC score to specific categories based on the breadth of therapy: 0–4 for narrow-spectrum, 5–9 for narrow-to-moderate, 10–14 for moderate, 15–19 for broad, and $\geq 20$ for very broad-spectrum therapy, adapted from previous literature that categorizes antibiotic coverage in five-point increments [18].

## Statistical analysis

Descriptive statistics were used to summarize the frequency, distribution, and characteristics of the isolated organisms at the three hospitals, focusing on the prevalence of MDRO. Comparative analyses of nominal variables were conducted using Pearson's chi-squared test and Fisher's exact test, and continuous variables were analyzed using the Kruskal-Wallis test. A p-value $< 0.05$ was considered statistically significant. IBM SPSS Statistics (version 27.0; IBM Corp., Armonk, NY, USA) was used to perform all calculations and statistical analyses.

## Ethical approval

This study was approved by the institutional review board of the three hospitals (approval No. Seoul Medical Center, 2023-08-001; Ajou University Hospital, AJOUIRB-DB-2023-498; Sanggye Paik Hospital, 2023-08-016). As this was a retrospective study using medical records, no additional information was obtained from the patients, and the need for documentation of informed consent was waived. Patient data were accessed for research purpose on October 18th, 2023: the informatics department of each hospital processed the primary data and

distributed to the authors anonymized data with patient numbers, which could be used for patient identification either during or after data collection when necessary.

## Results

A total of 3,064 patients, comprising 860, 1,855, and 349 patients from hospital S, hospital A, and hospital P, respectively, were enrolled in this study. The median age of the patients was 72 years (IQR: 61–82 years), 45.9% were female, and they significantly differed in age (p<0.001) and gender (p<0.001) between the institutions. Among the three institutions, the frequency of malignancy as an underlying disease showed significant differences (p<0.001) between the three institutions, with hospital A being the highest (20.3%), followed by hospital P (19.5%) and hospital S (3.7%). More than two bacterial species were isolated in 8% of cases. However, in the remaining 8%, multiple bacterial species were concurrently detected. Medical records revealed that 26.7% of the patients received antibiotics before blood culture collection, and hospital P exhibited a particularly high incidence of pre-sampling antibiotic administration at 71.3% with statistical significance (p<0.001) (Table 1).

A total of 3,397 bacterial isolates were identified from the 3,064 individuals: 905, 2,070, and 422 from hospital S, hospital A, and hospital P, respectively. Among the isolates, *Escherichia coli* was the most frequently identified bacterium in hospitals S and A, whereas *Staphylococcus epidermidis* was the most frequently detected bacterium in hospital P (S1 Table).

Among the 1,551 gram-positive and 1,527 gram-negative species isolated from blood cultures, MDRO was found in 893 patients, with the isolates comprised of 170 IRGN, 714 MRS, and 65 VRE. MRS constituted the majority of the MDRO in each hospital. Hospital P exhibited the highest proportion of MDRO across all categories (Fig 1).

Notably, 66% of all gram-positive organisms were methicillin-resistant, and approximately 40% were gentamicin-resistant. Resistance to teicoplanin and linezolid was reported in 7% and 1%, respectively (Table 2).

In gram-negative bacteria, resistance to piperacillin-tazobactam, imipenem, amikacin, and tigecycline was reported in 20%, 12%, 3%, and 12%, respectively (Table 3). Strains that were

**Table 1. Demographic and clinical characteristics of the multicenter study population.**

| Characteristic | Hospital S (n = 860) | Hospital A (n = 1,855) | Hospital P (n = 349) | Total (n = 3,064) | P-value |
|---|---|---|---|---|---|
| Female, number (%) | 473 (55.0%) | 800 (43.1%) | 148 (42.4%) | 1,421 (45.9%) | p < 0.001[a] |
| Age, median [IQR] | 78 [68–85] | 69 [58–80] | 74 [65–82] | 72 [61–82] | p < 0.001 |
| Underlying disease, number (%) | | | | | |
| • Malignancy | 32 (3.7%) | 376 (20.3%) | 68 (19.5%) | 476 (15.4%) | p < 0.001[a] |
| • HIV | 0 (0.0%) | 2 (0.1%) | 0 (0.0%) | 2 (0.1%) | p = 1.000[b] |
| • Organ transplantation | 0 (0.0%) | 35 (1.9%) | 0 (0.0%) | 35 (1.1%) | p < 0.001[b] |
| Number of isolated species from blood culture (BC), number (%) | | | | | |
| • 1 Species | 816 (94.9%) | 1,672 (90.1%) | 332 (95.1%) | 2,820 (92%) | p < 0.001[b] |
| • 2 Species | 43 (5.0%) | 157 (8.5%) | 16 (4.6%) | 216 (7%) | |
| • 3 Species | 1 (0.1%) | 21 (1.1%) | 1 (0.3%) | 23 (0.8%) | |
| • 4 Species | 0 | 4 (0.2%) | 0 (0%) | 4 (0.1%) | |
| • 5 Species | 0 | 1 (0.1%) | 0 (0%) | 1 (0%) | |
| Antibiotic administration before blood culture sampling, number (%) | 213 (24.8%) | 356 (19.2%) | 249 (71.3%) | 818 (26.7%) | p < 0.001[a] |

[a] Pearson's chi-squared test was performed to determine statistical significance.

[b] Fisher's exact test was performed due to the low expected frequencies.

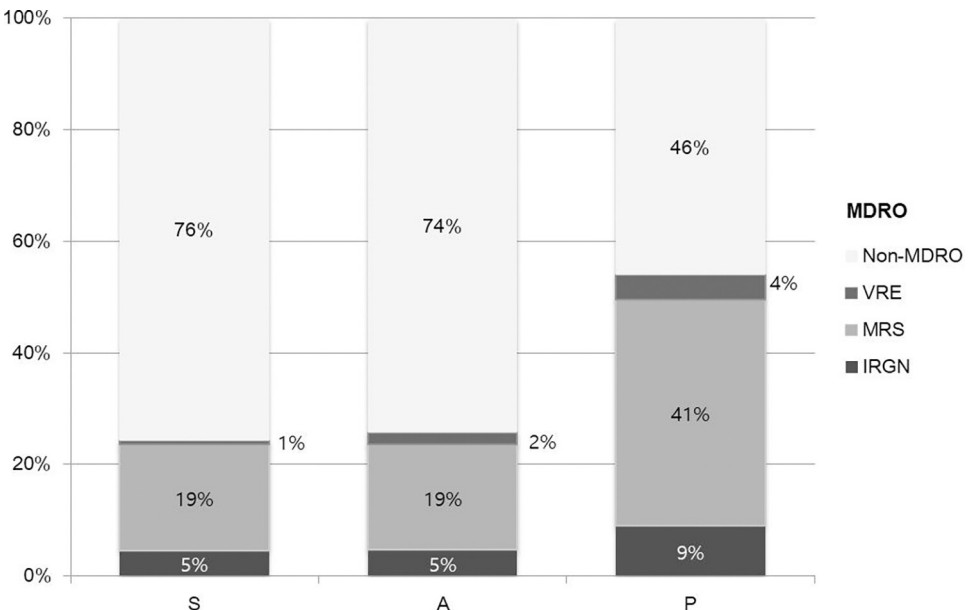

**Fig 1. Proportions of multidrug-resistant organisms (MDROs) within blood culture isolates from the three hospitals (S, A, and P).** The bar graph shows the prevalence of imipenem-resistant gram-negative bacteria (IRGN), methicillin-resistant *Staphylococcus* spp. (MRS), and vancomycin-resistant enterococci (VRE) across the hospitals. MRS (light gray) is the predominant MDRO, followed by IRGN (dark gray) and VRE (medium gray). Notably, Hospital P demonstrated the highest incidence of each MDRO type, whereas Hospitals S and A showed similar MDRO profiles.

not susceptible to imipenem yet were susceptible to ertapenem constituted 4.3% (55/1,293) of all Enterobacterales isolates. Most of these strains were identified as *Proteus* species, with 77.1% (37/48) of all *Proteus* species isolates demonstrating resistance to imipenem and

**Table 2. Antimicrobial susceptibility profiles of gram-positive organisms isolated from blood cultures.**

| Antimicrobial | Overall susceptibility | Susceptibility in MRS[a] | Susceptibility in VRE[b] |
|---|---|---|---|
| Ampicillin | 59.8% (257/430) | | 7.7% (5/65) |
| Ampicillin/Sulbactam | 47.8% (130/272) | | 7.7% (5/65) |
| Cefotaxime | 94.9% (150/158) | | |
| Ceftriaxone | 78.2% (158/202) | | |
| Ciprofloxacin | 50.9% (604/1186) | 33.3% (238/714) | 0.0% (0/23) |
| Clindamycin | 57.3% (762/1331) | 46.0% (328/713) | 0.0% (0/23) |
| Erythromycin | 42.3% (641/1515) | 32.1% (229/714) | 4.6% (3/65) |
| Gentamicin | 58.4% (627/1074) | 46.0% (329/715) | |
| Linezolid | 99.1% (1484/1497) | 99.4% (711/715) | 93.8% (61/65) |
| Oxacillin | 33.3% (359/1077) | 0.0% (0/717) | |
| Quinupristin/Dalfopristin | 94.0% (657/699) | 100.0% (325/325) | 86.2% (50/58) |
| Rifampicin | 92.3% (1000/1083) | 88.9% (635/714) | |
| Teicoplanin | 92.3% (1246/1350) | 92.9% (666/717) | 33.8% (22/65) |
| Tetracycline | 79.7% (1061/1331) | 78.7% (562/714) | 100.0% (23/23) |
| Trimethoprim/Sulfamethoxazole | 75.5% (903/1196) | 74.2% (532/717) | 0.0% (0/23) |
| Vancomycin | 95.6% (1483/1551) | 100.0% (717/717) | 0.0% (0/65) |

[a] Methicillin-resistant *Staphylococcus* spp.

[b] Vancomycin-resistant enterococci.

**Table 3. Antimicrobial susceptibility profiles of gram-negative organisms isolated from blood cultures.**

| Antimicrobial | Overall susceptibility | Susceptibility in IRGN[a] |
|---|---|---|
| Amikacin | 97.0% (1406/1450) | 76.4% (107/140) |
| Amoxicillin/Clavulanic Acid | 68.9% (902/1310) | 21.7% (20/92) |
| Ampicillin | 20.7% (275/1331) | 11.5% (10/87) |
| Cefepime | 69.7% (1036/1486) | 32.2% (55/171) |
| Cefotaxime | 60.9% (919/1509) | 20.6% (35/170) |
| Ceftazidime | 66.6% (1011/1519) | 27.5% (47/171) |
| Ciprofloxacin | 59.7% (903/1513) | 26.9% (46/171) |
| Colistin | 93.5% (157/168) | 94.9% (74/78) |
| Ertapenem | 97.4% (1259/1293) | 64.0% (55/86) |
| Gentamicin | 77.4% (1150/1485) | 40.6% (69/170) |
| Imipenem | 88.5% (1314/1485) | 0.0% (0/171) |
| Minocycline | 66.1% (119/180) | 87.3% (69/79) |
| Piperacillin/Tazobactam | 79.4% (565/712) | 21.8% (24/110) |
| Tigecycline | 87.8% (1298/1479) | 53.5% (91/170) |
| Trimethoprim/Sulfamethoxazole | 66.7% (1018/1527) | 36.5% (62/170) |

[a]Imipenem-resistant gram-negative bacteria.

susceptibility to ertapenem. In particular, 13 and 42 gram-positive isolates were resistant to linezolid and quinupristin/dalfopristin, respectively, whereas 44 and 181 gram-negative isolates were resistant to amikacin and tigecycline, respectively. The in-hospital mortality rate among patients with MDRO bacteremia (MDRO group) was 27.3% (244/893), which was significantly higher compared to 17.5% (379/2,171) mortality rate observed in patients with bacteremia caused by organisms other than MDRO (non-MDRO group), with statistical significance (p<0.001).

For the ASC score analysis, medical records pertaining to antimicrobial prescriptions were utilized for 2,400 patients, following the exclusion of individuals lost to follow-up on day R+2. Interestingly, the proportion of ASC scores ≥ 15, indicating broad- or extremely broad-spectrum antibiotic coverage, was not significantly different between the MDRO and non-MDRO groups at B0 yet significantly decreased in the non-MDRO group at R+2 (p<0.001, Table 4).

**Table 4. Distribution of patients according to the antibiotic spectrum coverage (ASC) score grade.**

| Antibiotic spectrum | ASC grade (score) | Empirical treatment (B0) | | Tailored treatment (R+2) | |
|---|---|---|---|---|---|
| | | Non-MDRO | MDRO | Non-MDRO | MDRO |
| Narrow | Grade 1 (0~4) | 27.2% (460) | 27.9% (198) | 45% (761) | 42.6% (302) |
| Narrow-to-moderate | Grade 2 (5~9) | 32.3% (546) | 21.9% (155) *** | 26.5% (448) | 20.5% (145) ** |
| Moderate | Grade 3 (10~14) | 22.2% (375) | 28.9% (205) *** | 19.2% (324) | 17.1% (121) |
| Broad | Grade 4 (15~19) | 9.1% (154) | 10.4% (74) | 6.4% (109) | 13.7% (97) *** |
| Very broad | Grade 5 (≥20) | 9.2% (156) | 10.9% (77) | 2.9% (49) | 6.2% (44) *** |
| Total | | 100% (1691) | 100% (709) | 100% (1691) | 100% (709) |

Pearson's chi-squared test was performed for each ASC grade, and statistically significant differences in MDRO values compared to non-MDRO are indicated as *
(p<0.05)

** (p<0.01), and

*** (p<0.001).

[a]B0, the day when blood culture was performed.

[b]R+2, the second day after the blood culture results were obtained.

## Discussion

This retrospective study demonstrated the significant mortality burden of bacteremia and the emergence of MDRO as major pathogens associated with bacteremia. According to the Global Antimicrobial Resistance and Use Surveillance Program in Korea (Kor-GLASS), which collects and annually reports AMR data across nine tertiary hospitals across Korea [19], a high prevalence of multidrug resistance were observed among blood-isolated organisms in large-scale hospitals in Korea. In 2022, the susceptibility rate of Staphylococcus aureus to cefoxitin was 54.3%, and *Enterococcus faecium* to vancomycin was 64.9%. Carbepenem susceptibility among blood-isolated Enterobacterales has also steadily declined since 2019 (imipenem-susceptible *Escherichia coli* and *Klebsiella pneumoniae*- 99.2% in 2019 to 94.0% in 2022) [20]. These data corroborate the substantial prevalence of MDRO among BSI in Korea. Interestingly, the susceptibility to imipenem was lower than that of ertapenem among the gram-negative isolates in this study, mainly for two reasons. First, susceptibility to ertapenem was not tested in non-fermenting gram-negative species, leading to the potential exclusion of carbapenem-resistant organisms in this group. Second, most IRGNs that demonstrated susceptibility to ertapenem were identified as *Proteus* and *Morganella* species, which are known to have elevated minimal inhibitory concentration to imipenem by mechanisms other than by the production of carbapenemases. Adverse clinical outcomes associated with MDRO infections compared with non-MDRO infections have been widely reported. Previous studies have reported that the proportion of MDRO in overall bacteremia ranges from 28–31%, with an overall mortality of 25–40% [21,22]. A recent Southern European cohort study reported 23.8% mortality for infections caused by carbapenem-resistant Enterobacterales

(CRE) compared to 10.6% for non-CRE infections, with a similar trend being observed in a subgroup analysis of bloodstream infections caused by CRE [23]. Other studies and meta-analyses have reported worse outcomes for infections caused by CRE [24,25], MRSA [26], and VRE [27,28] than their susceptible counterparts. Therefore, the lack of optimal therapeutic agents against MDRO warrants concern for these organisms.

The proportion of patients with underlying malignancy- at greater risk of BSI- was lower in hospital S. Hospital P reported a higher proportion of MRS and IRGN than the other two institutions. The proportions of *E. coli* and *K. pneumoniae* were also significantly lower at hospital P. Differences in the proportion of MDRO in bacteremia between institutions have also been previously reported, with MDRO being more common in long-term care facility-acquired sepsis. This phenomenon may have been influenced by the relatively poor compliance with disinfection in long-term care facilities and by the infection type [29]. Differences between institutions emphasize the effect of local epidemiology on the antimicrobial susceptibility profile pattern of a medical institution, which should be considered when designing empiric antibiotic regimens and antibiotic policies.

Although molecular markers associated with AMR have been extensively investigated and are examined in some clinical microbiology laboratories, current available commercial molecular panels may not provide sufficient information to direct appropriate antimicrobial choice for bacteremia. For example, most PCR-based molecular-based panels target only genes with established association to specific AMR, such as *mecA/C* in *Staphylococcus aureus*, *vanA/B* in *Enterococcus* species, and beta-lactamase genes [30,31]. The proportion of resistance against other antibiotic classes such as aminoglycoside, glycylcycline, and oxazolidinone, among commonly encountered pathogens was relatively low in this study. Nevertheless, it is present and may contribute to additional mortality due to ineffective empiric antimicrobial therapy or the virulence of the organisms themselves, as demonstrated by the previously mentioned analyses of CRE, MRSA, and VRE infections. Rapid detection of such antimicrobial resistance is

necessary to better prepare physicians for these infections. At least one United States Food and Drug Administration (FDA)-approved PCR-based panel with AMR gene detection extending to aminoglycoside, fluoroquinolone, trimethoprim-sulfamethoxazole, and polymyxin is available and has reported > 94% positive percent agreement and negative percent agreement compared with the composite reference standard [32]. However, most panels with such extended capabilities appear limited to research use only [33], likely because of a lack of clinical data, high equipment costs, or other issues. Furthermore, the correlation between the presence of AMR genes and phenotypic resistance to antibiotics has not yet been adequately established in many cases, and further studies are required to accurately predict of phenotypic resistance associated with AMR genes. However, some resistance determinants, such as *optrA* for linezolid and *tetA* for tigecycline, are known to be associated with phenotypic resistance [34,35]. While treatment options for MDRO are extremely limited, earlier detection of these resistance mechanisms would prevent physicians from empirically prescribing non-susceptible antibiotics and would help them determine the prognosis of bacteremia.

The proportion of patients receiving broad or extremely broad-spectrum antibiotics was not significantly different between the non-MDRO and MDRO groups at the empirical treatment stage yet significantly decreased in the non-MDRO group at the post-culture reporting stage. This implies that physicians appropriately adjust antibiotic regimens according to the susceptibility pattern of the organisms and that a more rapid report on the presence of antibiotic resistance may lead to earlier antibiotic de-escalation. Several studies have reported that the adoption of rapid molecular rapid diagnostic tests is associated with decreased time to effective therapy [8,36], suggesting that physicians react to microbiological reports on a timely basis. One study examined the impact of a real-time PCR assay detecting methicillin-resistant and methicillin-susceptible *S. aureus* in tissue specimens from skin and soft tissue infections on antibiotic prescription and found 48 cases where antibiotic modification was recommended based on the results of molecular diagnostics and two-thirds of the cases accepted the recommendations. The findings of this study suggest a possible role for rapid molecular tests in antimicrobial stewardship [37]. A recent survey also found that intensivists were willing to incorporate molecular diagnostics results when deciding on antibiotic regimens for pneumonia [38]. Although the potential suboptimal interpretation of molecular diagnostics reports is a concern [39], faster confirmation of antibiotic resistance through molecular diagnostics in bacteremia may encourage more rapid antibiotic de-escalation.

This study has several limitations. First, this study was conducted retrospectively and was therefore susceptible to selection bias. Second, the susceptibility profile in this study was obtained from commercialized breakpoint panels and was not confirmed by broth microdilution methods. However, this method allowed analysis of susceptibility profiles for a wide range of antimicrobial agents. Third, the three participating hospitals in this study were geographically clustered, and a single hospital accounted for nearly 60% of the positive blood cultures included in this study, which may also have been a source of selection bias. Nonetheless, the study cohort is comprised of patients from three hospitals with distinct capabilities and patient populations, and the results may more accurately reflect the overall picture of bacteremia among large-scale hospitals located in Seoul and the surrounding metropolitan areas. Finally, this study may have underestimated underlying diseases and comorbid conditions because these data were extracted only from the main diagnosis codes assigned to the patients. However, the proportion of underlying diseases was described by simply comparing the patient profiles across the three institutions, and we did not aim to analyze the data associated with BSI in this specific patient group.

In conclusion, this study found that resistance to last-resort antimicrobials, such as linezolid, tigecycline, and aminoglycoside, is emerging in Korea and that higher mortality is

observed in patients with MDRO bacteremia. In addition, antibiotic prescribers escalate or de-escalate antibiotic regimens according to the reported antimicrobial susceptibility profile on a timely basis. Therefore, developing and incorporating molecular diagnostics that detect a wider range of antimicrobial resistance than what is currently available may facilitate rapidly tailored antimicrobial treatment.

## Supporting information

**S1 Table. Prevalence of microbial isolates in blood cultures across three hospital settings.** (DOCX)

## Author Contributions

**Conceptualization:** Kuenyoul Park.

**Formal analysis:** Jin Sae Yoo, Kuenyoul Park.

**Writing – original draft:** Jin Sae Yoo, Hui-Jin Yu.

**Writing – review & editing:** Kuenyoul Park, Wee Gyo Lee, Bo-Moon Shin.

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
