## [Decision Letter · Decision Letter 0]

19 Jun 2024

PONE-D-24-17809Emergence of resistance to last-resort antimicrobials in bacteremia pathogens: A multicenter analysis of bloodstream pathogens in Korea.PLOS ONE

Dear Dr. Park,

Thank you for submitting your manuscript to PLOS ONE. After careful consideration, we feel that it has merit but does not fully meet PLOS ONE’s publication criteria as it currently stands. Therefore, we invite you to submit a revised version of the manuscript that addresses the points raised during the review process.

We look forward to receiving your revised manuscript.

Kind regards,

Mabel Kamweli Aworh, DVM, MPH, PhD. FCVSN

Academic Editor

PLOS ONE

2. In the online submission form, you indicated that [Restrictions apply to the availability of the data described in this study, and so are not publicly available. Subsets of datasets generated and analyzed during this study are available from the corresponding author on reasonable request.]. 

Additional Editor Comments (if provided):

Reviewers' comments:

Reviewer's Responses to Questions

**Comments to the Author**

1. Is the manuscript technically sound, and do the data support the conclusions?

Reviewer #1: Yes

Reviewer #2: Yes

2. Has the statistical analysis been performed appropriately and rigorously? 

Reviewer #1: Yes

Reviewer #2: No

3. Have the authors made all data underlying the findings in their manuscript fully available?

Reviewer #1: Yes

Reviewer #2: No

4. Is the manuscript presented in an intelligible fashion and written in standard English?

Reviewer #1: Yes

Reviewer #2: Yes

5. Review Comments to the Author

Reviewer #1: Comments and Suggestions for Authors

I read the manuscript with great interest. Here are some comments, which may help the authors to improve the manuscript:

Title: I suggest the “bacteremia pathogen” in the title should be replaced with “bacteremia patients”

Abstract: Please define MDRO at its first mention. This applies to all others throughout the paper.

Methods: For easy understanding, I suggest the methods should be grouped under sub-headings.

For example;

Lines 92 – 104 should be under sub-heading: Study design and procedures

Lines 105 – 119 should be under the sub-heading: Classification of Antibiotic Resistance

Lines 120 – 125 should be under the sub-heading: Statistical analysis

Lines 126 – 133 should be under the sub-heading: Ethical approval

Reviewer #2: REVIEWER’S COMMENTS

Title: Emergence of resistance to last-resort antimicrobials in bacteremia pathogens: A multicenter analysis of bloodstream pathogens in Korea

Summary

This is an important research work which brings to fore the profile of AMR associated with blood stream infection in Seoul, South Korea. It is a retrospective multi-centre study conducted within a year period across three health facilities (two secondary and one tertiary health facility) with the aim of conducting a large-scale analysis that enables adequate review of the microbiological and clinical characteristics of patients diagnosed with bacteremia. The objectives of the study and rationale for the study were well highlighted. The problem statements and gap in knowledge were also well outlined. Reference to previous literature were satisfactory.

The authors described their methods broadly which is worrisome for reproducibility. To summarize, the authors reviewed positive blood culture results to identify bacteria isolates, conducted antimicrobial susceptibility testing and compared the pattern of empirical treatment and tailored treatments with respect to their spectrum of coverage and MDRO. They went further to assess the underlying pathology and in-hospital mortality of the patients. However, there are important questions to answer. The authors conducted this study in only urban hospitals which have implications for generalizability to a more heterogenous population. The study sites should be sufficiently described to put the study in context. The sampling method and study population were not adequately described. The study population appears to be skewed to the elderly based on the median age group despite the fact that bacteremia can occurs in any age group. It is important that the authors clearly outline the inclusion and exclusion criteria of the study population.

Although, the variables of interest were well elucidated, there was no mention of the study instrument which has implications for standardization being a multi-centre study. Were the samples analyzed separately in each facility or pooled to a centre and were there clear Laboratory protocols developed? What were the other quality assurance processes for the laboratory component of this study?

The research findings provide evidence of antimicrobial resistance to last resort antibiotics. Analysis done are mostly descriptive, authors should consider doing some inferential statistics or if done to be reflected in the tables. In addition, given the comparative nature of the study with respect to facilities, statistically significant differences should also be highlighted (asterix) on the tables where available. It is also important that the authors explain how 3397 bacteria isolates were isolated from 3064 blood patients, having mentioned in the methods, that they focused solely on the first organism isolated from each positive blood culture per patient. Figure 1, should add up to 100% for each health facility (showcasing MDROs and non-MDROs even if the MDROs are disaggregated). The variables in figure 2 are best presented using a table to compare the ASC scores between Empirical and Tailored treatment and highlighting the level of statistical significance.

The authors may also wish to review the manuscript for typographical and grammatical errors beginning with the abstract, line 27-28. Also, line 50-57 (The sentence is too long, authors to consider splitting into 2 or 3 sentences). Rephrase sentence in line 77-80. The discussion in line 177-183 is a bit confusing.

Overall, the study findings highlight the need for rapid diagnostic infrastructure with capacity for wide antimicrobial resistance detection in order to reduce the use of empirical antibiotics and enhance timely prescription of tailored treatment which is critical to slow down the emergence of multidrug resistant organisms.

I personally think this is a well written manuscript, the findings are noteworthy and should be used to drive a strong public health response in South Korea to address the issues of antimicrobial resistance. I recommend that the manuscript be published after the outlined corrections are effected.

6. PLOS authors have the option to publish the peer review history of their article (what does this mean?). If published, this will include your full peer review and any attached files.

Reviewer #1: No

Reviewer #2: **Yes: **DR. JENNY ADONORELI MOMOH

---

## [Author Response · Author response to Decision Letter 0]

11 Jul 2024

July 11, 2024

Dr. Emily Chenette

Editor-in-Chief

PLOS ONE

Dear Dr. Calderaro:

On behalf of the authors, I would like to thank you and the reviewers for the valuable and insightful comments they have provided regarding our submitted manuscript, “Emergence of resistance to last-resort antimicrobials in bacteremia pathogens: A multicenter analysis of bloodstream pathogens in Korea” (manuscript ID: PONE-D-24-17809).

We have carefully reviewed the comments and have revised the manuscript accordingly. Our responses are provided in a point-by-point manner, and changes to the manuscript are shown in yellow highlight, which is also visible through the Track Changes function.

We want to reiterate that this manuscript has not been published or presented elsewhere in part or in entirety and is not under consideration by another journal. We have read and understood your journal's policies, and we believe that neither the manuscript nor the study violates any of these.

Thank you for your consideration. I look forward to hearing from you. 

Sincerely,

Kuenyoul Park, MD, PhD. 

Department of Laboratory Medicine, Sanggye Paik Hospital, School of Medicine, Inje University, 1342, Dongil-ro, Nowon-gu, Seoul 01757, Korea. 

Tel: +82-2-950-4898

Fax: +82-2-950-1244

Email: kuenyoul.park@gmail.com 

Response to Reviewer 1

Thank you for reviewing our manuscript. We appreciate the reviewer's thorough analysis and comments. We have considered the reviewer's points as much as possible to improve the quality of our research and paper. We have answered each of your points below.

1. Title: I suggest the “bacteremia pathogen” in the title should be replaced with “bacteremia patients”

è Response: Thank you for this insightful suggestion. We have revised the title as follows:

“Emergence of resistance to last-resort antimicrobials in bacteremia pathogens: A multicenter analysis of bloodstream pathogens in Korea” → “Emergence of resistance to last-resort antimicrobials among bacteremia patients: A multicenter analysis of bloodstream pathogens in Korea”

2. Abstract: Please define MDRO at its first mention. This applies to all others throughout the paper.

è Response: Response: Following your suggestion, we defined MDRO at its first mention in the abstract.

“Among these, 949 isolates obtained from 893 patients were classified as multidrug-resistant organisms (MDRO), including 170 imipenem-resistant gram-negative bacteria, 714 methicillin-resistant staphylococci, and 65 vancomycin-resistant enterococci.”

3. Methods: For easy understanding, I suggest the methods should be grouped under sub-headings.

For example;

Lines 92 – 104 should be under sub-heading: Study design and procedures

Lines 105 – 119 should be under the sub-heading: Classification of Antibiotic Resistance

Lines 120 – 125 should be under the sub-heading: Statistical analysis

Lines 126 – 133 should be under the sub-heading: Ethical approval

è Response: Thank you for your helpful suggestion. We added subheadings as recommended.

Response to Reviewer 2

Thank you for reviewing our manuscript. We appreciate the reviewer's thorough analysis and comments. We have carefully considered the reviewer's points and have incorporated them to enhance the quality of our research and paper. Below, we have addressed each of your points.

The authors conducted this study in only urban hospitals which have implications for generalizability to a more heterogeneous population. The study sites should be sufficiently described to put the study in context. 

è Response: Thank you for your insightful suggestion. The three participating hospitals in this study are geographically clustered; however, each hospital's characteristics and functional capacity are distinct. Hospital A is a university-affiliated tertiary-care hospital with high patient severity, and hospital P is a secondary-care hospital with a larger volume of oncology patients compared to other secondary-care hospitals. The proportion of cancer among patients enrolled from these two hospitals is approximately 20% as shown in Table 1. While hospital S is also a secondary-care hospital, it is a municipal hospital that carries out public projects and serves patients of various socioeconomic status: elderly patients, who generally have lower income compared to the general population, particularly utilize hospital S due to its affordability. We did attempt to incorporate as much heterogeneity to our study population by including several hospitals with distinct characteristics, yet we also acknowledge that the hospitals are not geographically diverse and added this point as a limitation of our study under “Discussion” as follows:

"Third, the three participating hospitals in this study were geographically clustered, and a single hospital accounted for nearly 60% of the positive blood cultures included in this study, which may also have been a source of selection bias.” 

The sampling method and study population were not adequately described. The study population appears to be skewed to the elderly based on the median age group despite the fact that bacteremia can occurs in any age group. It is important that the authors clearly outline the inclusion and exclusion criteria of the study population.

è Response: All hospitalized patients with positive blood cultures during the study period. We clarified the study duration and inclusion/exclusion criteria under “Methods” as follows:

“This retrospective observational study was conducted using the medical records and blood culture results of patients admitted to three distinct hospitals from July 2022 to June 2023: a public secondary hospital (S), a university-affiliated tertiary hospital (A), and a university-affiliated secondary hospital (P). All hospitalized patients with positive blood cultures during the study period were enrolled. Data were collected only for the first organism isolated from positive blood culture of each patient’s duration of admission: in case two or more organisms were simultaneously identified on first blood culture, all were included.”

è We also agree that the age distribution appears skewed toward the elderly population. However, the high median age (72 years) and interquartile age (61 to 82 years) of this study are more likely attributable to the high risk of bacterial infection associated with elderly population (e.g., reduced efficacy of immune system and comorbidities such as diabetes).

Although, the variables of interest were well elucidated, there was no mention of the study instrument which has implications for standardization being a multi-centre study. Were the samples analyzed separately in each facility or pooled to a centre and were there clear Laboratory protocols developed? What were the other quality assurance processes for the laboratory component of this study? 

è Response: We clarified the testing process and the quality control policies under “Methods”:

“Bacterial identification and AST were performed in the laboratory of the patient’s admitting hospital. The laboratories of all three participating hospitals were accredited for internal and external quality assurance programs by the Laboratory Medicine Foundation of Korea and Korean Association of External Quality Assessment Service during the study period."

The research findings provide evidence of antimicrobial resistance to last resort antibiotics. Analysis done are mostly descriptive, authors should consider doing some inferential statistics or if done to be reflected in the tables. In addition, given the comparative nature of the study with respect to facilities, statistically significant differences should also be highlighted (asterix) on the tables where available. 

è Response: We thank you for sharing this perspective. The demographic and clinical characteristics were compared between the three institutions using the Kruskal-Wallis test for continuous variables and the chi-square or Fisher’s exact test for categorical data. The statistical analysis methods and results were added to the text as follows:

“Comparative analyses of nominal variables were conducted using Pearson's chi-squared test and Fisher's exact test, while continuous variables were analyzed using the Kruskal-Wallis test.”

“The median age of the patients was 72 years (IQR: 61–82 years), 45.9% were female, and there were statistically significant differences in age (p<0.001) and gender (p<0.001) between the institutions.”

“Among the three institutions, the frequency of malignancy as an underlying disease showed statistically significant differences (p<0.001) between the three institutions, being highest in hospital A (20.3%), followed by hospital P (19.5%) and hospital S (3.7%).”

It is also important that the authors explain how 3397 bacteria isolates were isolated from 3064 blood patients, having mentioned in the methods, that they focused solely on the first organism isolated from each positive blood culture per patient. 

è Response: If the same bacterial pathogen was repeatedly identified from the same patient within the patient’s admission duration, only the first identification result was retained, and the rest were excluded to avoid repeated inclusion. However, as noted in Table 1, two or more distinct bloodstream pathogens were simultaneously identified from a single patient within a single admission in approximately 8% of the patients, in which case all distinct organisms were included. We clarified this point in “Methods”: 

“Data were collected only for the first organism isolated from positive blood culture of each patient’s duration of admission: in case two or more organisms were simultaneously identified on first blood culture, all were included.”

Figure 1, should add up to 100% for each health facility (showcasing MDROs and non-MDROs even if the MDROs are disaggregated). 

è Response: We revised Figure 1 as recommended. 

The variables in figure 2 are best presented using a table to compare the ASC scores between Empirical and Tailored treatment and highlighting the level of statistical significance.

è Response: We converted Figure 2 into Table 4, and statistical significance were denoted with asterix. We also incorporated these changes in “Results” as follows:

“Interestingly, the proportion of ASC scores ≥ 15, indicating broad- or very broad- spectrum antibiotic coverage, was not significantly different between the MDRO and non-MDRO groups at B0, but significantly decreased in non-MDRO group at R+2 (p<0.001, Table 4).”

The authors may also wish to review the manuscript for typographical and grammatical errors beginning with the abstract, line 27-28. 

è Response: Thank you for pointing out this detail. We corrected the error as follows: 

“Antibiotic spectrum coverage (ASC) scores were calculated on the day the blood culture was performed (B0) and on two days after the blood culture results were reported obtained (R+2).” → “Antibiotic spectrum coverage (ASC) scores were calculated on the day the blood culture was performed (B0) and on two days after the blood culture results were reported (R+2).” 

Also, line 50-57 (The sentence is too long, authors to consider splitting into 2 or 3 sentences). 

è Response: We divided the sentence as follows: 

“Bacteria harboring resistance to several classes of antimicrobial agents are commonly referred to as multidrug-resistant organisms (MDRO).”

“In 2017, the World Health Organization (WHO) listed several MDRO as “priority pathogens” requiring novel therapeutic agents, including vancomycin-resistant enterococci (VRE), methicillin-resistant Staphylococcus aureus (MRSA), carbapenem-resistant Acinetobacter baumannii, carbapenem-resistant Pseudomonas aeruginosa, and carbapenem-resistant/extended beta-lactamase (ESBL)-producing Enterobacterales.”

Rephrase sentence in line 77-80. 

è Response: Thank you for this suggestion. We rephrased this as follows: 

“Recently, T2resistance (T2Biosystems, Lexington, MA, USA) has become available for research use and one study reported 84.6% detection rate of carbapenem resistance genes in blood-culture confirmed carbapenem-resistant gram-negative BSI with a turnaround time of 3 to 5 hours” → “Recently, T2resistance (T2Biosystems, Lexington, MA, USA) has become available for research use, and one study reported 84.6% detection rate of carbapenem resistance genes in blood culture-confirmed Gram-negative BSI within 3 to 5 hours.” 

The discussion in line 177-183 is a bit confusing. 

è Response: We rewrote this as follows in the revised manuscript: 

“According to the Global Antimicrobial Resistance and Use Surveillance Program in Korea (Kor-GLASS) data, which collects and annually reports AMR data across nine tertiary hospitals across Korea, there was high prevalence of multidrug-resistance among blood-isolated organisms among large-scale hospitals in Korea. In 2022, susceptibility rate of blood-isolated Staphylococcus aureus to cefoxitin was 54.3%, and susceptibility rate of Enterococcus faecium to vancomycin was 64.9%. Carbepenem susceptibility among blood-isolated Enterobacterales has also steadily declined since 2019 (imipenem-susceptible Escherichia coli and Klebsiella pneumoniae- 99.2% in 2019 to 94.0% in 2022).”

---

## [Decision Letter · Decision Letter 1]

20 Aug 2024

Emergence of resistance to last-resort antimicrobials in bacteremia patients: A multicenter analysis of bloodstream pathogens in Korea.

PONE-D-24-17809R1

Dear Dr. Park,

We’re pleased to inform you that your manuscript has been judged scientifically suitable for publication and will be formally accepted for publication once it meets all outstanding technical requirements.

Kind regards,

Mabel Kamweli Aworh, DVM, MPH, PhD. FCVSN

Academic Editor

PLOS ONE

Additional Editor Comments (optional):

Reviewers' comments:

Reviewer's Responses to Questions

**Comments to the Author**

1. If the authors have adequately addressed your comments raised in a previous round of review and you feel that this manuscript is now acceptable for publication, you may indicate that here to bypass the “Comments to the Author” section, enter your conflict of interest statement in the “Confidential to Editor” section, and submit your "Accept" recommendation.

Reviewer #1: All comments have been addressed

Reviewer #2: All comments have been addressed

2. Is the manuscript technically sound, and do the data support the conclusions?

Reviewer #1: (No Response)

Reviewer #2: Yes

3. Has the statistical analysis been performed appropriately and rigorously? 

Reviewer #1: (No Response)

Reviewer #2: Yes

4. Have the authors made all data underlying the findings in their manuscript fully available?

Reviewer #1: (No Response)

Reviewer #2: No

5. Is the manuscript presented in an intelligible fashion and written in standard English?

Reviewer #1: (No Response)

Reviewer #2: Yes

6. Review Comments to the Author

Reviewer #1: (No Response)

Reviewer #2: Title: Title: Emergence of resistance to last-resort antimicrobials in bacteremia pathogens: A multicenter analysis of bloodstream pathogens in Korea

Following the review of the updated manuscript, I think that the authors have to a very great extent addressed the queries raised. They have addressed the issues around heterogeneity of study sites and generalization of study findings. They have also provided more clarity into their methods and laboratory procedures to enable for reproducibility. Further review of their results have been done which now transcends beyond descriptive analysis to include inferential statistics, giving more weight to the findings.

I am satisfied with the current update on the manuscript as the authors have taken the time to address point by point all issues raised. I therefore recommend that the manuscript be published.

7. PLOS authors have the option to publish the peer review history of their article (what does this mean?). If published, this will include your full peer review and any attached files.

Reviewer #1: No

Reviewer #2: No

---

## [Editor Report · Acceptance letter]

23 Aug 2024

PONE-D-24-17809R1 

PLOS ONE

Dear Dr. Park, 

I'm pleased to inform you that your manuscript has been deemed suitable for publication in PLOS ONE. Congratulations! Your manuscript is now being handed over to our production team.

Kind regards, 

on behalf of

Dr. Mabel Kamweli Aworh 

Academic Editor

PLOS ONE